# A Novel Trace-Level Ammonia Gas Sensing Based on Flexible PAni-CoFe_2_O_4_ Nanocomposite Film at Room Temperature

**DOI:** 10.3390/polym13183077

**Published:** 2021-09-12

**Authors:** Rima D. Alharthy, Ahmed Saleh

**Affiliations:** 1Department of Chemistry, Science and Arts College, Rabigh Campus, King Abdulaziz University, Jeddah 21577, Saudi Arabia; iaaalharte@kau.edu.sa; 2Science and Technology Center of Excellence (STCE), Cairo 3066, Egypt

**Keywords:** polyaniline, cobalt ferrite, ammonia gas sensor, flexible, nanocomposite

## Abstract

In this study, we developed a new chemi-resistive, flexible and selective ammonia (NH_3_) gas sensor. The sensor was prepared by depositing thin film of polyaniline-cobalt ferrite (PAni-CoFe_2_O_4_) nanocomposite on flexible polyethylene terephthalate (PET) through an in situ chemical oxidative polymerization method. The prepared PAni-CoFe_2_O_4_ nanocomposite and flexible PET-PAni-CoFe_2_O_4_ sensor were evaluated for their thermal stability, surface morphology and materials composition. The response to NH_3_ gas of the developed sensor was examined thoroughly in the range of 1–50 ppm at room temperature. The sensor with 50 wt% CoFe_2_O_4_ NPs content showed an optimum selectivity to NH_3_ molecules, with a 118.3% response towards 50 ppm in 24.3 s response time. Furthermore, the sensor showed good reproducibility, ultra-low detection limit (25 ppb) and excellent flexibility. In addition, the relative humidity effect on the sensor performance was investigated. Consequently, the flexible PET-PAni-CoFe_2_O_4_ sensor is a promising candidate for trace-level on-site sensing of NH_3_ in wearable electronic or portable devices.

## 1. Introduction 

Ammonia (NH_3_) is a pungent, toxic, colorless, water-soluble and flammable gas produced worldwide in large quantities of more than 200 million tons per annul. It plays a vital role in all life forms and is one of the major industrial raw materials in chemicals production facilities such as agriculture, refrigeration technology, food, fertilizers and medical facilities [1,2]. According to Occupational Safety and Health Administration (OSHA) and the Agency for Toxic Substances and Disease Registry (ATSDR), ammonia is hazardous gas and its presence in atmosphere even at very low concentrations of 50 ppm could irritate the respiratory tract, skin, nose and throat of children and/or adults. It also has a bad impact on the environment and can cause lung damage or even death at high concentration levels beyond 500 ppm [3]. Therefore, a low-cost, sensitive, stable, ambient temperature, and reliable ammonia detection sensor is imperative. Recently, conductive conjugated polymers on flexible substrate are in fashion as sensing materials for trace-level detection of ammonia owing to their lightweight, flexible, and portable nature [4]. These type of ammonia sensors are widely reported due to their simple synthesis, ambient temperature sensitivity and low cost processing [5,6,7,8]. In this context, polyaniline (PAni) is one of the most significant conducting polymers used for ammonia sensing because of its high reactivity and facile synthesis [9] along with its reversible doping/dedoping property [10], excellent electrical properties, unique redox characteristics and adjustable sensing at ambient temperature. However, pure PAni-based sensors have certain associated caveats, such as limited sensing efficiency in context of response, selectivity, and long response/recovery time [11]. Recently, researchers focused more on PAni-based nanocomposites using functionalized carbon-based materials and metal oxide semiconductors [12,13,14,15,16,17] that resulted in a significant performance improvement in PAni-based ammonia sensors.

The current tendency in producing trace-level NH_3_ gas sensors is to prepare flexible substrate sensors based on PAni as conducting polymers due to their lightweight, portable, and flexible properties. Y. Zhang et al. [18] prepared an NH_3_ sensor based on polyaniline/SrGe_4_O_9_ nanocomposite on Polyimide (PI) substrate by in situ chemical oxidation polymerization technique. The sensor revealed excellent response time (24 s), good flexibility and reproducibility. Q. Wu et al. [4] utilized porous polyvinylidene fluoride (PVDF) as aflexible template for PAni composed with graphene (GP) with 10% response towards 0.1 ppm NH_3_ and response time (46 s). J. Ma et al. [19] modified PET fibers by ethylenediamine (EDA) to expose amino groups and adhere to carboxyl groups of MWCNTs and coated by PAni, the sensor showed 117% response towards 50 ppm NH_3_ with response time (47 s) and cost-effective flexible substrate modification.

Cobalt ferrite (CoFe_2_O_4_) is an n-type semiconductor with band gap of 1.76 eV [20]. This material showed promising magnetization properties [21,22], including high remanence magnetization and coercivity. It also has shown high chemical stability, cost-effectiveness, and shape versatility when produced at high temperatures. Cobalt ferrites have been applied in many technological fields, such as ferrofluids, catalysis, electronics, cancer treatment [23] and chemi-resistive sensors [24]. It has an inverse spinel structure in which, Fe^3+^ ions are distributed between octahedral and tetrahedral sites and Co^2+^ ions are in octahedral sites. In oxidation reactions, CoFe_2_O_4_ has significant catalytic properties due to the high mobility of oxygen ion at the film surface and thus is highly preferred for the gas sensing applications [25].

Herein, we report a thin film of PAni-CoFe_2_O_4_ nanocomposites with different concentrations of CoFe_2_O_4_ NPs deposited on a flexible PET substrate by using in situ chemical oxidative polymerization technique. The prepared sensor is subsequently applied for room temperature detection NH_3_ gas. Several gas sensing parameters, such as selectivity, response at different gas concentrations, reproducibility, response/recovery times, flexibility, and low detection limit were studied. 

## 2. Experimental 

### 2.1. Materials

Cobalt (II) nitratehexahydrate [Co(NO_3_)_2_·6H_2_O, ≥99% Fluka, Germany], Iron(III) nitratenonahydrate (Fe(NO_3_)_3_·9H_2_O, ≥98%, from Sigma-Aldrich), citric acid as fuel (99.6%, Acros), Aniline monomer (C_6_H_7_N, Merck, Germany, 99.5%), ammonium persulfate, APS ((NH_4_)_2_S_2_O_8_, Acros, Belgium, 98%) as an oxidant, hydrochloric acid (37%) and ammonia solution (35%) (Fisher Scientific, Belgium) were used as received. Polyethylene terephthalate (PET) film with dimensions of 7 cm × 3 cm with ±80 μm thickness was used without any further treatment. DI water was used for the synthesis of CoFe_2_O_4_ NPs and the polyaniline nanocomposites.

### 2.2. Fabrication of Flexible PET-PAni-CoFe_2_O_4_ Sensor Films 

CoFe_2_O_4_ nanoparticles (NPs) were prepared by sol–gel combustion technique as reported by L. E. Caldeira et al. [26] as following; (Co(NO_3_)_2_·6H_2_O), (Fe(NO_3_)_3_·9H_2_O) and citric acid were dissolved in 20 mL DI with 1:2:3 molar ratio. The mixture was stirred and heated at 85 °C for 1 h. The gel was dried for 24 h at 110 °C to remove the water. The xerogel was sintered in muffle furnace for 6 h at 750 °C. In situ chemical oxidative polymerization technique was used to prepare PAni-CoFe_2_O_4_ nanocomposite films on flexible PET substrates by polymerization of C_6_H_7_N monomer in dilute hydrochloric acid using (NH_4_)_2_S_2_O_8_ as an oxidant. The method leads to the deposition of PAni-CoFe_2_O_4_ thin film on flexible substrate (in this work, PET). Before use, the PET films were cleaned by immersing in boiling acetone and then in isopropyl alcohol followed by drying for 1 h at 70 °C. The experimental procedure is as follows: PET films were immersed vertically in solution of 0.2 M C_6_H_7_N monomer dissolved in 1 M HCl at 0–5 °C. The mixture was stirred using a mechanical stirrer supplied with Teflon rod to avoid any agglomeration. After ~1 h, 0.1 M (NH_4_)_2_S_2_O_8_ was drop-wise added to the above mixture with continuous stirring and kept at these conditions for further 14 h. Finally, a green emeraldine salt precipitate of PAni was formed. The as-synthesized CoFe_2_O_4_ NPs (10, 30 and 50 wt%) were sonicated and added to the C_6_H_7_N for the preparation of hybrid nanocomposite on the PET film. The flexible film was filched from the PAni solution, washed with DI water and finally placed in an oven at 60 °C for 1 h. The screen-printing technique was used to print the silver-integrated electrode onto the surface of the film to obtain the flexible gas sensor film. The scheme of film formation is depicted in Figure 1. 

### 2.3. Characterization and Gas Detection Measurements

As-prepared samples were investigated for their chemical structure by Fourier transform infrared spectroscopy (FT-IR) (Thermo Nicolet Avatar 370). Structural study was conducted by XRD (ARL X’TRA Powder Diffractometer, Thermo Scientific) at a speed of 5° per min, 2θ scan range 5–80° with λ 1.5406 Å. Surface topography and surface roughness average (Ra) for flexible sensors were measured by a profilometer with contact stylus tracing (KLA Tencor™ P-7, Milpitas, California, USA). The diameter of the diamond tip was 2.4  mm with accuracy 1  mm/s. Measurements were performed in three different positions of each film, Ra Mean values were calculated for each sample. The surface morphology was carried out by FE-SEM (Quanta FEG 250, Waltham, MA, USA). CoFe_2_O_4_ particle size was investigated using HR-TEM (JEOL-2100, Tokyo, Japan). The glass transitions (*T*_g_) of the flexible sensors were carried out using a (TA Instruments DSC Q4000, New Castle, DE, USA). The thermal stability of samples was monitored by TGA using a (TA Instrument TGA Q500, New Castle, DE, USA) with nitrogen purge flow 50 mL per min at 25–900 °C with a ramp rate of 10 °C per min. PAni-CoFe_2_O_4_ (50%) nanocomposite elemental composition was evaluated by XPS (K-ALPHA, Themo Fisher Scientific, Waltham, MA, USA). CoFe_2_O_4_ NPs hysteresis loop parameters were evaluated by a Vibrating Sample Magnetometer (VSM) (Lakeshore-7410, Westerville, OH, USA). The gas sensing properties of the sensors were evaluated at 40% relative humidity and room temperature by a homemade chamber as shown in Figure 2. The chamber is attached with a real-time acquisition system to measure the resistance. Prior to introducing NH_3_ gas in to the test chamber, the sensor film was stabilized for 20 min to obtain the stable baseline in dry air, then a gas stream in the ambient temperature with various concentrations in the range of 1–50 ppm was injected to evaluate the sensors responses NH3 and selectivity test of interfering gas molecules (CO_2_, C_2_H_5_OH and CH_3_OH) at 50 ppm. A mass flow controller (MFC 300) (was utilized to expose a constant flux of 50 cm^3^ min^−1^ from dry air to various injected target gas). To desorb NH_3_ after each test, the sensor was flushed with dry air. The % response of the sensor is expressed by the following equation:(1)$$\mathrm{Response}\;(\%)=\frac{R_{g}-R_{o}}{R_{o}}\times 100$$
where *R_o_* and *R_g_* are the measured resistances of air and tested gas, respectively [27,28]. The recovery time (T_rec_) and response time (T_res_) were measured as the times taken by the film sensor to reach 90% of the resistance change [29,30]. The humidity effect on PET-PAni-CoFe_2_O_4_ sensor was detected using humidity chamber (Cincinnati Sub Zero—CSZ) by replacing the test gas chamber in the environmental chamber and adjusting the desired value of RH for 20 min till equilibrium before testing.

## 3. Results and Discussion

### 3.1. Magnetic Properties of CoFe_2_O_4_ NPs

The calcination temperature effect on the magnetic parameters of the precursor was recorded by VSM at room temperature, Figure 3 displays the M-H hysteresis loop for the CoFe_2_O_4_ NPs. The hysteresis curve shows ferrimagnetic performance of spinel CoFe_2_O_4_ nanocrystals. The coercive field (Hc), saturation magnetization (Ms), remanence magnetization (Mr), and the squareness (R) value (Mr/Ms), were 1523 (Oe), 68.5 (emu/g), 29.5 (emu/g), and 0.43, respectively. The value of Ms is lower than the observed bulk (74.08 emu/g) [31]; this may be attributed to the modified cationic distribution and the disorder of the nanoparticle’s surface. This is an indication of the excellent magnetic properties of CoFe_2_O_4_.

### 3.2. FT-IR Analysis

FT-IR spectra of PAni, CoFe_2_O_4_ NPs, and PAni-CoFe_2_O_4_ (50%) nanocomposite are presented in Figure 4. The spectrum of PAni have a peak at 797 cm^−1^ due to C—H bending of aromatic ring (out of plane) [32]. The absorption peaks at 1131 cm^−1^ and 1293 cm^−1^ are attributed to the C—N stretching of quinoid and benzenoid ring, respectively [33]. The peaks at 1479 cm^−1^ and 1563 cm^−1^ represent thestretching vibration C=C of benzenoid and quinoid structure, respectively [34]. The peak observed at 2411 cm^−1^ is due to C=NH^+^ [35]. The PAni spectrum has a prominent peak at 3417 cm^−1^ for aromatic amine —N-H stretching while the peaks at 2919 and 2850 cm^−1^ are attributed to —C-H stretching. The above-mentioned typical peaks confirm the synthesis of PAni in protonated state. CoFe_2_O_4_ have strong peaks at 580 and 493 cm^−1^ which represent the vibrations (ν) of M-O (Metal (M)=Fe, Co) symmetric stretching in tetrahedral and octahedral sites of CoFe_2_O_4_ [36,37]. The peak at 1633 cm^−1^ corresponds to bending vibration of H-O-H [23]. The peak at 3432 cm^−1^ can be inferred to stretching vibration of O-H group at the surface of NPs due to surface water [38]. It is important to note that that the peak of CoFe_2_O_4_ in the spectrum of PAni-CoFe_2_O_4_ (50%) nanocomposite shifted to higher wavenumbers of 584 cm^−1^ that indicates the strong interaction between metal ions of CoFe_2_O_4_ and nitrogen atoms of PAni due to hydrogen bonding.

### 3.3. XRD Analysis

The XRD pattern of CoFe_2_O_4_ NPs, PAni and PAni-CoFe_2_O_4_ (50%) nanocomposite powder is presented in Figure 5. CoFe_2_O_4_ NPs showed reflection planes namely (220), (311), (222), (400), (422), (511) and (440). The observed peaks matched with (JCPDS 00-002-1045, space group Fd3m, space group number no 227) for reflections of cubic spinel structure. The the average crystallite size as obtained by the Scherrer formula was found to be 41 nm.

For PAni, two broad peaks at 20 and 25° are related to the amorphous structure of PAni [39]. In PAni-CoFe_2_O_4_ (50%), no diffraction peak for PAni was noticed in the pattern, which confirms the amorphous nature of PAni in its composites with metal oxides [40]. In the composite synthesis, the ammonium persulfate as an oxidative initiator proceeded on the CoFe_2_O_4_ NPs surface and encapsulated in PAni shell leading to the resistive effect of nanoparticles that hamper the crystallinity of PAni. The crystal form of CoFe_2_O_4_ NPs in PAni-CoFe_2_O_4_ composite pattern showed low crystallinity compared to CoFe_2_O_4_; the phenomenon is attributed to the coating effect and intermolecular interaction between surface of CoFe_2_O_4_ and conducting PAni [41]. 

### 3.4. Morphological Analysis (FE-SEM)

FE-SEM analysis was performed to visualize the morphology of the as-synthesized CoFe_2_O_4_ NPs, PET-PAni and PET-PAni-CoFe_2_O_4_ (50%) sensors, Figure 6. According to the results of FE-SEMSEM, the CoFe_2_O_4_ NPs, Figure 6a, have a tendency to agglomerate that is attributed to their ferrimagnetic properties [42] and sponge-like structure with porous morphology, which may have a great impact on the sensing properties. Despite the agglomeration and the blurred edges of CoFe_2_O_4_, the NPs are still uniform spheres with particle sizes of 70–80 nm. The larger grain size might be attributed to crystallites magnetic properties. 

A quantitative elemental EDS spectrum in Figure 6b is used to calculate the CoFe_2_O_4_ NPs composition, the molar ratio of Co, Fe, and O was found to be 1:1.96:3.94 that is fairly close to the theoretical values of 1:2:4, indicating the nominal composition and the stoichiometric proportion is maintained. It is clear from Figure 6c that PAni behaves in a similar way to a well-developed interconnected fiber with porous morphology in PET-PAni film. The surface morphology of PET-PAni-CoFe_2_O_4_ (50%) film was found in clusters, porous and rough surface, Figure 6d. Moreover, the agglomeration of CoFe_2_O_4_ NPs on the PAni surface was observed, which endorses literature reports showing that an increase in NPs loading in the matrix leads to coagulation [43].

### 3.5. TEM Analysis

TEM and SAED patterns were recorded for detailed insight into the microstructure of CoFe_2_O_4_ NPs (Figure 7). The particles have a spherical shape with particle size in the range of 40–80 nm (Figure 7a). Segregated as well as agglomerated particles are clearly visible which endorse FE-SEM data. Selected area electron diffraction (SAED) pattern shows white spot rings indicating the polycrystalline structure of CoFe_2_O_4_ NPs, Figure 7b. SAED results matched well with the XRD patterns. 

### 3.6. Roughness Measurements

Profilometer is one of the most commonly used devices for measuring the roughness (Ra) and 3D images for the nanocomposite. Generally, the roughness of the nanocomposite surface depends on the structure of the organic matrix and the content of the inorganic filler. The surface topography of PET-PAni shows a relatively uniform and smooth structure compared to PET-PAni-CoFe_2_O_4_ (50%) in Figure 8. The mean surface roughness of the PET-PAni and PET-PAni-CoFe_2_O_4_ (50%) samples were found to be 1.15 and 5.3 μm, respectively. This increase in Ra value lead to augmented polarity and surface area that results in extra growth sites and buildup of adhesion between PAni and CoFe_2_O_4_ NPs. 

### 3.7. Thermal Analyses

#### 3.7.1. Thermogravimetric Analysis (TGA)

TG and DTG thermograms of CoFe_2_O_4_, PAni, PAni-CoFe_2_O_4_ (50%), PET, PET-PAni and PET-PAni-CoFe_2_O_4_ (50%) are shown in Figure 9a,b. The degradation of CoFe_2_O_4_ occurred in two stages having an insignificant loss: the first step was up to 600 °C that is attributed to the loss of moisture, decarbonation and removal of hydroxyl group associated with the NPs (0.21%) while the second step at higher temperature was up to 900 °C (0.18%) [44]. The TGA thermogram confirms the high thermal stability of CoFe_2_O_4_ NPs. For pure PAni, there are three main stages of pyrolysis, the first between 30 to 140 °C that is attributed to loss of adsorbed water, unreacted monomer and free acid remaining after polymerization, 8.54% [45]. The second stage is found in a range of 140 to 325 °C due to the degradation of low molecular weight oligomers formed during the synthesis of PAni, 11.49% and residue 7.73% [46]. The third and last stage was in a temperature range 350 to 900 °C due to the structural breakdown of PAni molecules, 69.3% degradation and residue 14.5% [47]. PAni-CoFe_2_O_4_ (50%) nanocomposite have shown the same thermal degradation behavior as PAni, as they have a somewhat high thermal stability with a residue (33.2%). Moreover, the thermal stability of plastic films coated with nanocomposite thin film was also examined. The TG curves reveal that the decomposition of PET substrate occurred in one step in a temperature range of 300–520 °C, related to the random degradation of ester groups into carbonyl and vinyl ester which were subsequent converted to acetaldehyde by tautomerization effect [48] with a weight loss of 94.86% and char residue of 5.15% at 900 °C. The PET-PAni film showed typical behavior of PET but with more weight loss of 96.4% and residue of 3.6%, which may be attributed to the catalytic effect of PAni during the decomposition of PET film. The PET film coated with PAni-CoFe_2_O_4_ (50%) showed higher thermal stability compared to all other films with weight loss of 92.7% and residue of 7.3% due to the high thermal stability of CoFe_2_O_4_ embedded in the PET film.

#### 3.7.2. Differential Scanning Calorimetry (DSC) 

The interactions of PET film with PAni and CoFe_2_O_4_ during the in situ polymerization reaction were monitored, which were responsible for the nanocomposite phase transitions (see Figure 10). One-step DSC scanning is performed from ambient conditions to 300 °C. The glass transition (*T*_g_) of pristine PET, PET-PAni, and PET-PAni-CoFe_2_O_4_ (50%) were found to be 81 °C, 81.3 °C, and 82.7 °C, respectively. The slight increase in *T*_g_ may be attributed to the interaction between the PET substrate and nano-fillers that resulted in a reduction in the polymer segmental motion. To understand the effect of the coated nano-fillers on polymer structural characteristics, the polymer crystallinity was studied using the heat of fusion obtained from DSC thermograms and the degree of crystallinity, Xc%, was determined by using Equation (2) [49].
(2)$$\mathrm{Xc}\%=\frac{\Delta H}{\Delta H_{0}}\times 100$$
where ΔH is the PET melting enthalpy and ΔH_0_ is the 100% crystalline PET melting enthalpy, considered to be 140 jg^−1^ [50]. The PET crystallinity degree was found to be 30.6%, which shifted to higher values for PET-PAni and PET-PAni-CoFe_2_O_4_ (50%) to 32.5% and 35.7%, respectively. This increase in crystallinity may be attributed to the creation of segments of small chains, which are capable of crystallization and realigning easily, as indicated by TGA and FT-IR.

### 3.8. XPS Spectra

The chemical composition and the oxidation state of PAni-CoFe_2_O_4_ (50%) nanocomposite was elucidated by XPS measurements (Figure 11). The XPS spectrum confirms the presence of Co, Fe, N, O and C in the nanocomposite, Figure 11a. The core level of C1s (Figure 11b) deconvoluted into four distinct peaks at 283.8 eV, 284.7 eV and 287.3 eV, which belongs to —C=C, C—N, and C=O bands, respectively [51,52]. The peak at 285.7 eV is related to C—O [53]. Figure 11c shows the N1s’ core level and they fit well into three peaks located at 398.8 eV, 400.5 eV and 402.6 eV that can be attributed to quinoid immine (—N=), benzenoid amine (—NH—) and (N^+^) of PAni chain, respectively [54]. At lower binding energy, there are two peaks situated at 529.6 and 532 eV for O1s, Figure 11d. The peak at 529.6 eV confirms Co—O, while the peak at 532 eV is attributed to (OH) group and defect sites of oxygen [55]. In Figure 11e, the main peak Fe 2p_3/2_, with a satellite peak at 716.8 eV, is fitted into two peaks of Fe 2p_3/2_ and Fe 2p_1/2_ at binding energies 710.5 and 712.7, respectively. These peaks indicate the presence of Fe species in two different lattice positions, the peak at 710.3 eV from Fe^3+^ ions in octahedral sites while the peak at 712.7 eV from Fe^3+^ ions in tetrahedral sites [56]. Figure 11f shows two major peaks which can be attributed to Co 2p_3/2_ at 780 eV and Co 2p_1/2_ at 796.1 eV. Where Co 2p_3/2_ is well fitted into two peaks, at 779.8 eV for Co^2+^ in octahedral sites and 781.5 eV for Co^2+^ in tetrahedral sites with satellite peak at 786 eV, while Co 2p_1/2_ is well fitted into one peak at 796.2 eV for Co^2+^ in octahedral sites with satellite peak at 803.2 eV [57]. Thus, the analysis of XPS confirms the PAni-CoFe_2_O_4_ (50%) nanocomposite formation.

### 3.9. Gas Sensing Measurements

#### 3.9.1. Selectivity of PET-PAni-CoFe_2_O_4_ Film

Gas sensing selectivity is a critical parameter for evaluation any chemi-resistive gas sensor performance. Different gases were used to test the selectivity of PET-PAni-CoFe_2_O_4_ flexible film at 50 ppm of each gas and the bar charts of selectivity towards NH_3_, CO_2_, methanol and ethanol are shown in Figure 12a. The study clearly demonstrates the selectivity of the PET-PAni and PET-PAni-CoFe_2_O_4_ flexible films for NH_3_ compared to other gases. This selectivity may be due to the intense interactions between the sensing layers of flexible films and the adsorbed molecules of NH_3_ gas. Moreover, the PET-PAni-CoFe_2_O_4_ (50%) showed the highest value of response (118.3%) towards 50 ppm of NH_3_ compared to PET-PAni (10.16%). The response using the PET-PAni-CoFe_2_O_4_ (10%) of PET-PAni-CoFe_2_O_4_ was found to be 30.12% while it was 56.6% for the PET-PAni-CoFe_2_O_4_ (30%) (Figure 12b). The higher response value at PET-PAni-CoFe_2_O_4_ (50%) is mainly attributed to the porous structure [58], as noticed by FE-SEM. Hence, the PET-PAni-CoFe_2_O_4_ (50%) was selected for further analysis of its selectivity at room temperature sensing of NH_3_ gas.

#### 3.9.2. Response-Dependent Characteristics of PET-PAni-CoFe_2_O_4_ (50%) Film

The gas response time-dependent profile of PET-PAni-CoFe_2_O_4_ (50%) sensor towards 1–50 ppm of NH_3_ is shown in Figure 12c. The dynamic response profile revealed that the flexible sensor is sensitive to 1 ppm NH_3_ concentration with response of (6.1%). Moreover, the increase in NH_3_ gas concentration led to an increase in the sensor response that reaches its maximum response of 118.3% at 50 ppm NH_3_ gas. At higher NH_3_ concentration, the molecules of gas cover the sensor active sites and involved in surface interactions giving even higher response. Additionally, the relative humidity (RH) effect on the NH_3_ gas sensing of the PET-PAni-CoFe_2_O_4_ (50%) sensor was monitored for 50 ppm NH_3_ at different RH, and data is plotted in Figure 12d. The PET-PAni-CoFe_2_O_4_ (50%) sensor displayed a maximum response value of 124.8% at 20% RH, while the response decreased to 118.3% at 40% RH. Further increase in RH leads to a decrease in the sensor response, hence, the response is greatly affected by RH. Obviously, when the sensor film is subjected to high concentrations of RH, the molecules of water occupy some active sites of the sensor and overlap with the adsorbed molecules of the target gas that is probably reason of reduced response at higher RH [59,60]. The resistance dynamic change of PET-PAni-CoFe_2_O_4_ (50%) sensor when exposed to different concentrations of NH_3_ gas (1–50 ppm) is shown in Figure 12e.

#### 3.9.3. Reproducibility, Response-Recovery Times and Flexibility of the Sensor

The reproducibility and stability of the sensor are imperative characteristics to show its reliability. The reproducibility of PET-PAni-CoFe_2_O_4_ (50%) sensor was evaluated by repeating the exposure cycles for four times at 10 ppm NH_3_ and the response values are shown in Figure 13a. The PET-PAni-CoFe_2_O_4_ (50%) response is almost the same for four cycles. Thus, PET-PAni-CoFe_2_O_4_ (50%) sensor has excellent reproducibility and can be utilized repetitively for the room temperature and lower concentration detection of NH_3_ gas.

Figure 13b represents the times of response and recovery, which are measured from the dynamic response curves of PET-PAni-CoFe_2_O_4_ (50%) sensor at different gas concentrations shown in Figure 12c. The response and recovery times were inversely changed with NH_3_ gas concentrations, the sensor at 50 ppm of NH_3_ gas showed lowest response time (24.3 s). The effect is due to large availability of porous sites on the surface of the sensor for adsorption of gas requiring a short response time while the recovery time increased due to decrease in reactive species desorption rate after NH_3_ gas removal [58]. Furthermore, the response of PET-PAni-CoFe_2_O_4_ (50%) sensor toward very low NH_3_ concentrations (25–100 ppb) has also been evaluated and the response curve is shown in Figure 13c. It is clear that the developed PET-PAni-CoFe_2_O_4_ (50%) sensor is very sensitive to low concentrations and practically can detect as low as 25 ppb of NH_3_ gas that effectively meets the requirement of health and environmental issues. The flexibility of PET-PAni-CoFe_2_O_4_ (50%) sensor was evaluated after 500 bending cycles at 50 ppm NH_3_ and data is displayed in Figure 13d. An insignificant change in the sensor response (from 118.3% to 114.9%) is observed, indicating the excellent stability of PET-PAni-CoFe_2_O_4_ (50%) sensor after repeated cycles of bending. 

There are several reports in the literature for NH_3_ sensors. A comparison of the sensing performance of PET-PAni-CoFe_2_O_4_ (50%) developed in current study with the reported method is presented in Table 1. The PET-PAni-CoFe_2_O_4_ (50%) sensor developed in this study exhibits ultra-low detection limit, high response, and excellent flexibility, which shows its potential to be used in the portable NH_3_ sensing device development.

#### 3.9.4. Proposed Mechanism of Gas Sensing 

Protonation and deprotonation phenomenon resulting from NH_3_ gas adsorption and desorption is responsible for the response in pure PAni sensor. When NH_3_ gas is introduced to the sensor surface, the PAni emeraldine salt (ES) is converted to PAni emeraldine base (EB). Consequently, the PAni hole density decreases which leads to an increase in the resistance [64,65]. The gas sensing in the nanocomposite film predominantly depends on NH_3_ gas trapping that leads to the change in resistance. Figure 14 shows the schematic sensing model of PAni-CoFe_2_O_4_ flexible sensor at ambient. The n-type CoFe_2_O_4_ NPs were wrapped in the p-type PAni emeraldine salt to obtain porous nanocomposite due to integration of CoFe_2_O_4_ in the PAni matrix as observed in FE-SEM images which leads to formation active centers on the film surface with donor and acceptor states. Thus, it could be concluded that the p-n junctions are established at PAni-CoFe_2_O_4_ interface. When the sensor film is exposed to NH_3_, the molecules of gas would capture holes from —NH_2_— and =NH^+^— groups of PAni [66] and converts from emeraldine salt to emeraldine base as shown in Figure 14a, resulting in a decrease in PAni conductivity [14]. The adsorption of NH_3_ as a reducing gas leads to decrease in the concentration of PAni holes and p-n junctions depletion regions are widened as shown in shown in Figure 14b. The interaction between NH_3_ gas released free electrons, and hence neutralize the holes found on PAni surface due to electron-hole combination. This leads to a decrease in the concentration of hole and heterojunctions, as a result resistance of the nanocomposite sensor increases. 

## 4. Conclusions

In the present work, a flexible gas sensor composed of a thin film of PAni-CoFe_2_O_4_ nanocomposites on PET substrate by using simple in situ chemical oxidative polymerization technique with different CoFe_2_O_4_ NPs concentrations is reported. The developed sensor is subsequently applied successfully for room temperature NH_3_ gas sensing. The morphological analyses reveal porous structure the PAni-CoFe_2_O_4_ flexible film. The formation of PAni-CoFe_2_O_4_ nanocomposites confirmed by XPS analysis. The gas sensing data showed that the PET-PAni and PET-PAni-CoFe_2_O_4_ sensors have highest values of selectivity when exposed to NH_3_ gas at room temperature compared to other environmental gases. The PET-PAni-CoFe_2_O_4_ (50%) flexible sensor demonstrated a maximum response value of 118.3%, excellent response time of (24.3 s) at 50 ppm NH_3_ gas. The sensor has good reproducibility, ultra-low detection limit (25 ppb) and excellent flexibility with insignificant response change after repeated cycles of process and bending. Therefore, this paper highlights that the as-fabricated flexible PET-PAni-CoFe_2_O_4_ (50%) sensor provides a promising platform for trace-level NH_3_ detection of NH_3_ gas on chemicals production field and environmental testing. 

## Figures and Tables

**Figure 1 polymers-13-03077-f001:**
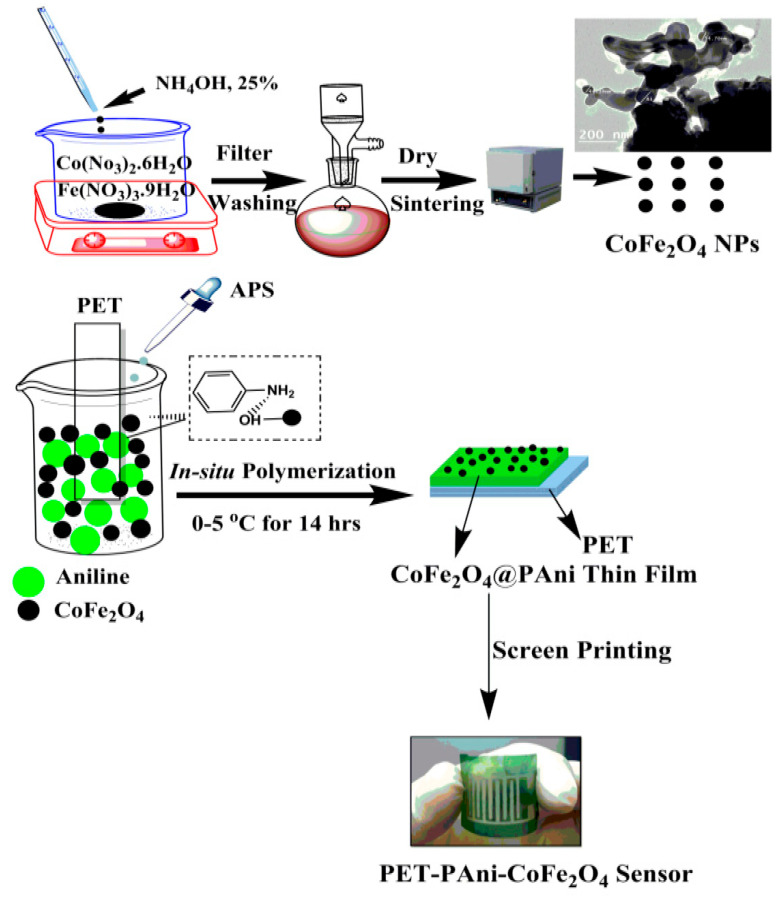
Flexible PET-PAni-CoFe_2_O_4_ nanocomposite film formation.

**Figure 2 polymers-13-03077-f002:**
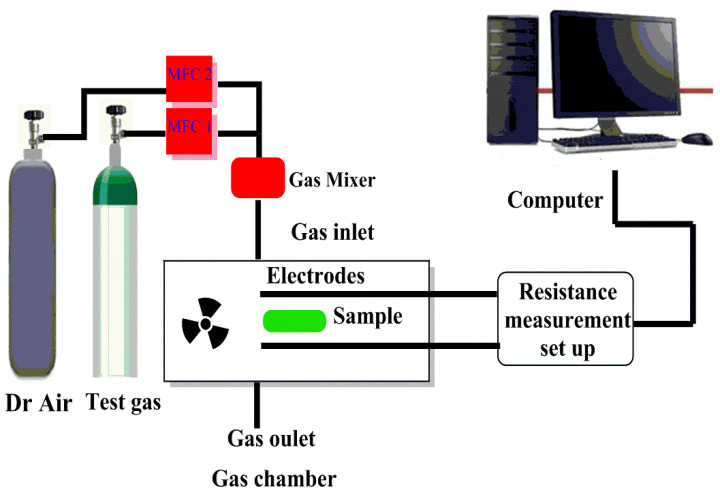
The gas sensing setup diagram.

**Figure 3 polymers-13-03077-f003:**
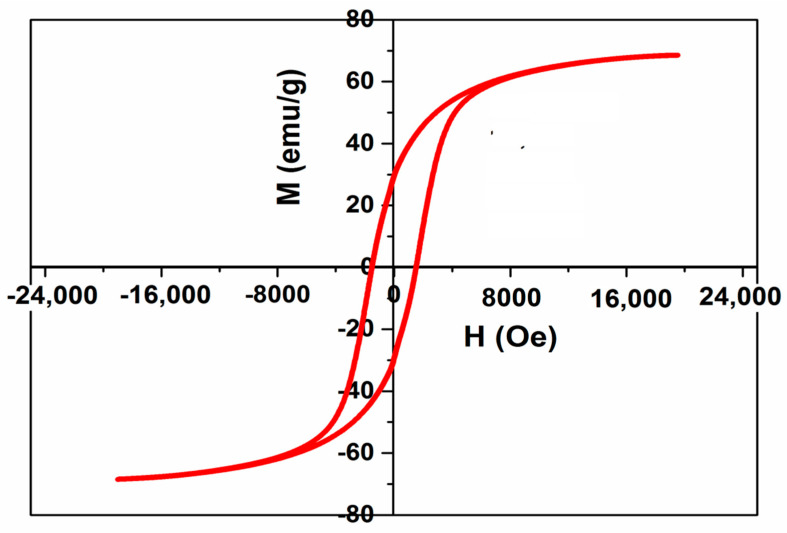
Hysteresis loop of CoFe_2_O_4_ sample.

**Figure 4 polymers-13-03077-f004:**
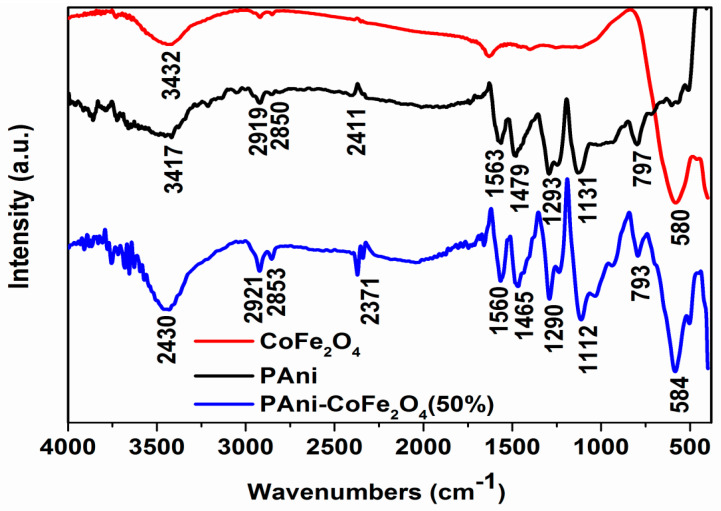
FT-IR of CoFe_2_O_4_, PAni and nanocomposite of PAni-CoFe_2_O_4_ (50%).

**Figure 5 polymers-13-03077-f005:**
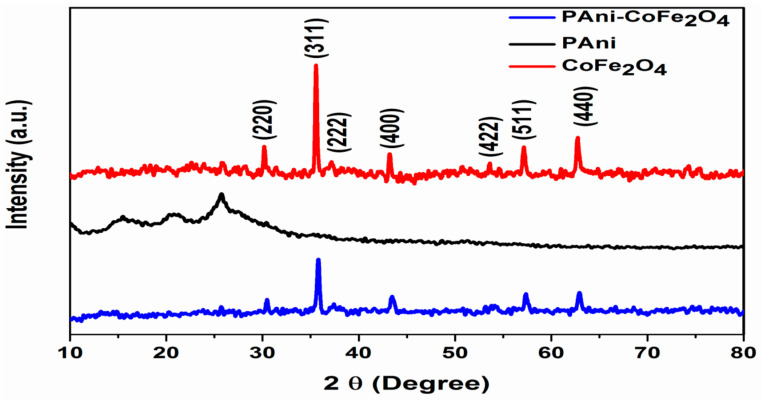
The XRD of CoFe_2_O_4_, PAni and nanocomposite of PAni-CoFe_2_O_4_ (50%).

**Figure 6 polymers-13-03077-f006:**
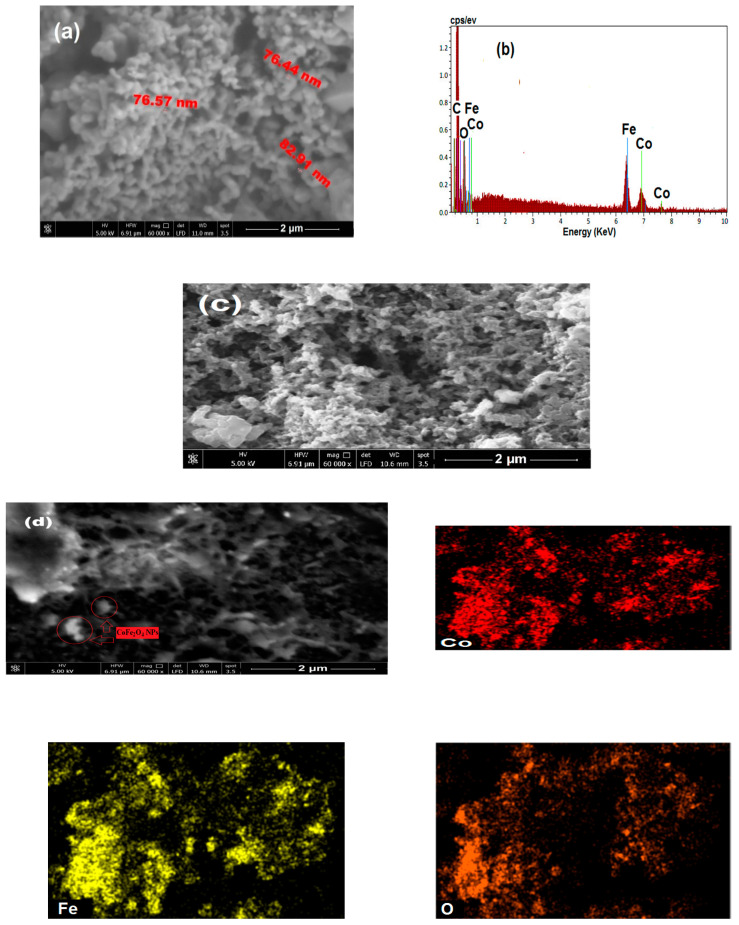
FE-SEM of (**a**) CoFe_2_O_4_, (**b**) EDS of CoFe_2_O_4_, (**c**) PET-PAni and (**d**) PET-PAni-CoFe_2_O_4_ (50%) sensor and its elemental mapping images of the Co (red), Fe (yellow) and O (brown) signals.

**Figure 7 polymers-13-03077-f007:**
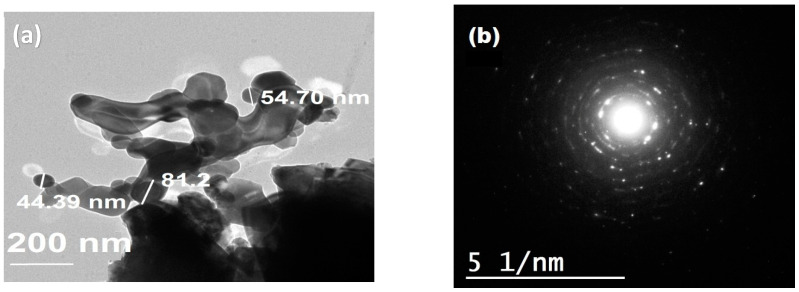
(**a**) TEM and (**b**) SAED pattern of CoFe_2_O_4_.

**Figure 8 polymers-13-03077-f008:**
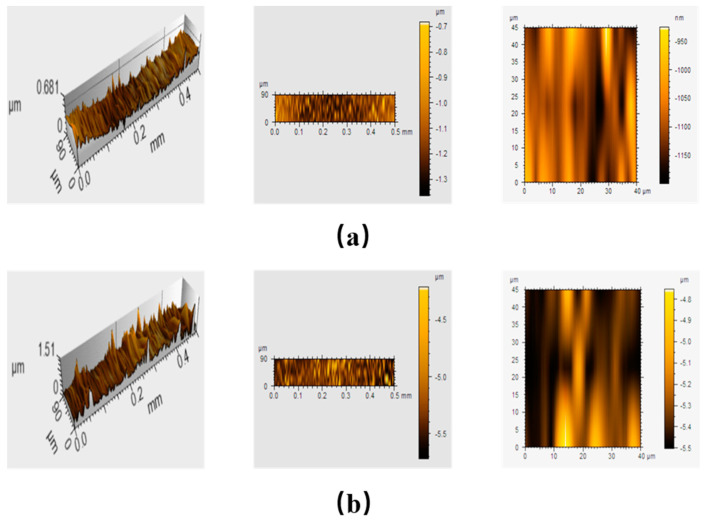
Surface structure images of (**a**) PET-PAni and (**b**) PET-PAni-CoFe_2_O_4_ (50%) measured by stylus profiler.

**Figure 9 polymers-13-03077-f009:**
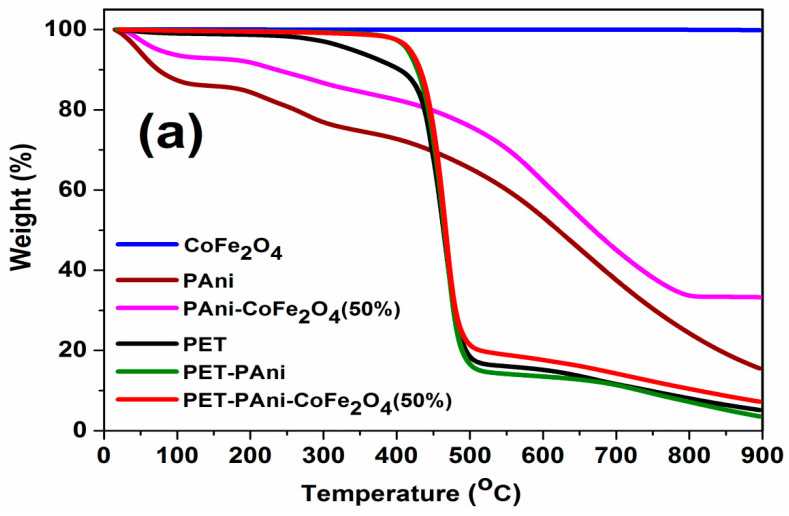
TG (**a**) and DTG curves (**b**) of CoFe_2_O_4_, PAni, PAni-CoFe_2_O_4_ (50%) nanocomposite, PET, PET-PAni and PET-PAni-CoFe_2_O_4_ (50%).

**Figure 10 polymers-13-03077-f010:**
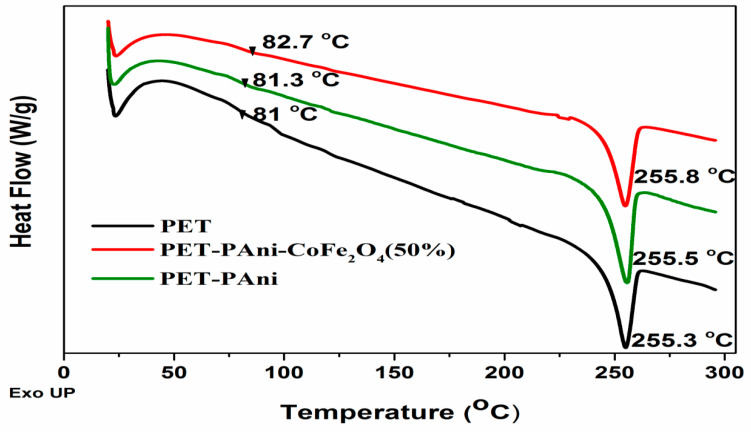
DSC thermograms of PET, PET-PAni and PET-PAni-CoFe_2_O_4_ (50%).

**Figure 11 polymers-13-03077-f011:**
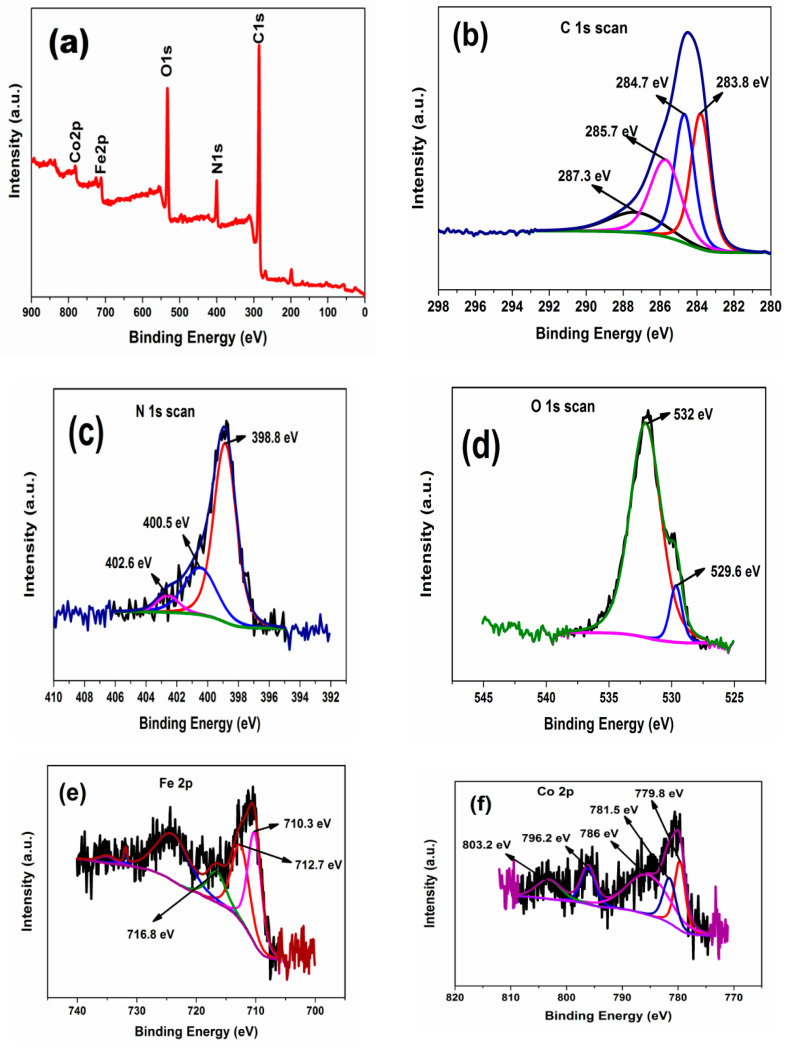
XPS of PAni-CoFe_2_O_4_ (50%) nanocomposite: (**a**) survey spectra, (**b**) C1s spectra, (**c**) N1s spectra, (**d**) O1s spectra, (**e**) Fe 2p, (**f**) Co 2p.

**Figure 12 polymers-13-03077-f012:**
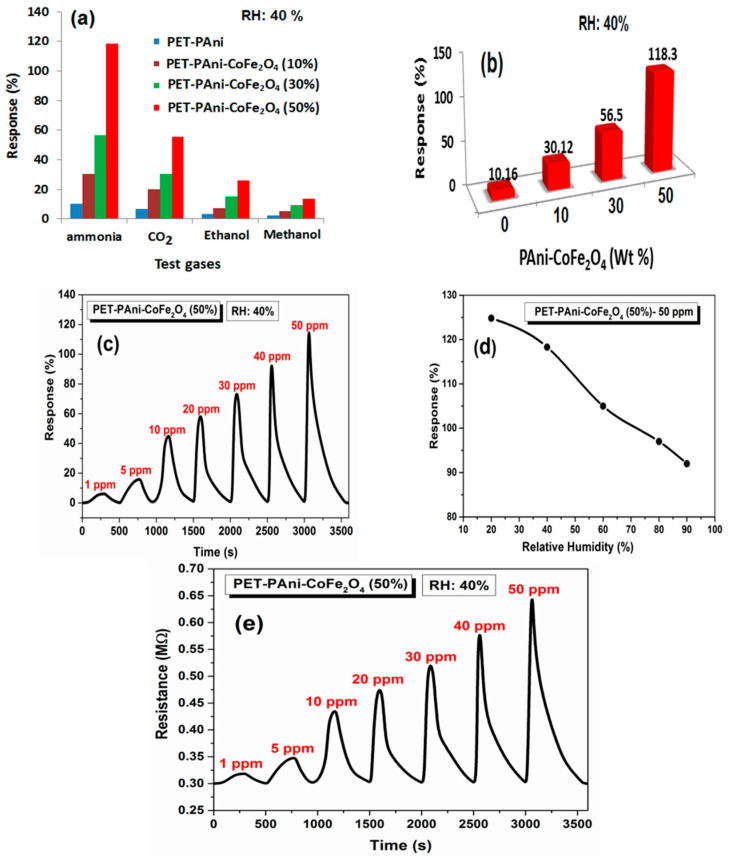
(**a**) Selectivity of PET-PAni and PET-PAni-CoFe_2_O_4_ films towards different test gases at 50 ppm, (**b**) Response of PET-PAni and PET-PAni-CoFe_2_O_4_ films towards 50 ppm NH_3_ gas, (**c**) Response of PET-PAni-CoFe_2_O_4_ (50%) to 1–50 ppm of NH_3_, (**d**) Humidity study of PET-PAni-CoFe_2_O_4_ (50%) at 50 ppm NH_3_ gas and (**e**) Resistance values of PET-PAni-CoFe_2_O_4_ (50%) to 1–50 ppm of NH_3_.

**Figure 13 polymers-13-03077-f013:**
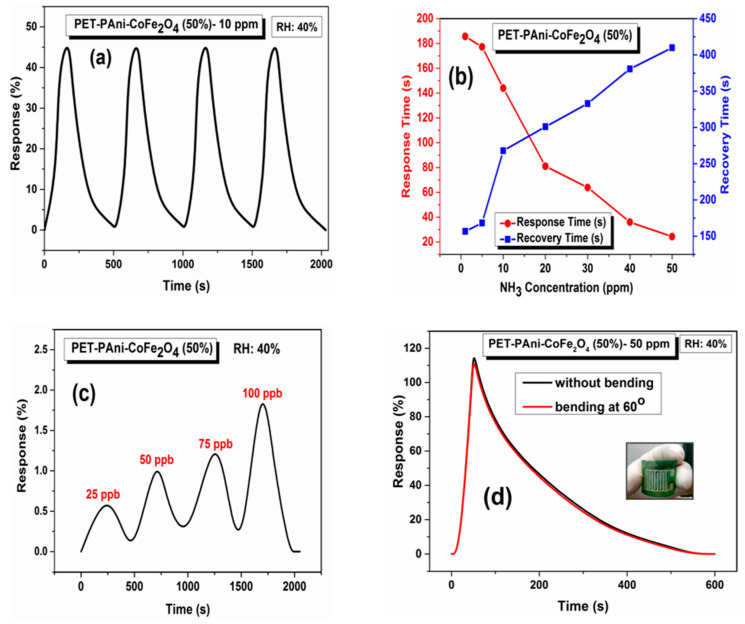
(**a**) Reproducibility of PET-PAni-CoFe_2_O_4_ (50%) to 10 ppm of NH_3_ gas, (**b**) Response and recovery time of PET-PAni-CoFe_2_O_4_ (50%) sensor, (**c**) detection limit of PET-PAni-CoFe_2_O_4_ (50%), (**d**) flexibility of PET-PAni-CoFe_2_O_4_ (50%) at bending angel 60° at 50 ppm of NH_3_ gas.

**Figure 14 polymers-13-03077-f014:**
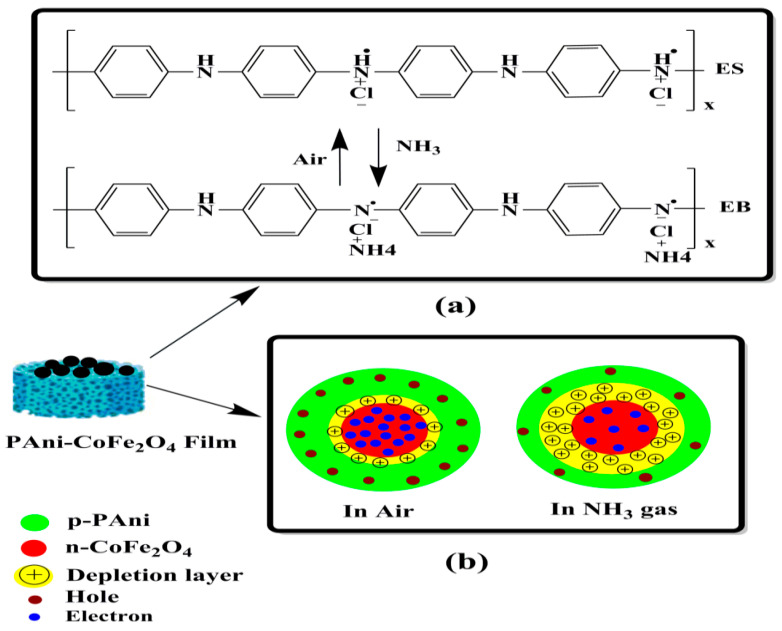
Schematic sensing model of PAni-CoFe_2_O_4_ flexible sensor, (**a**) the NH_3_ gas interaction with the PAni emeraldine salt, and (**b**) the depletion region changes of PET-PAni-CoFe_2_O_4_ in air and NH_3_ gas.

**Table 1 polymers-13-03077-t001:** The sensing properties comparison of PET-PAni-CoFe_2_O_4_ (50%) with other reported works.

Material	Substrate	Detection Limit	Response %	Response Time (s)	Flexibility	Ref.
PAni	PET	<5 ppm	26 (100 ppm)	33	-	[61]
S, N: GQDs/PAni	PET	1 ppm	42.3 (100 ppm)	115	The response at bending angle 80° was more than at 0°	[14]
MWCNT-PAni	PVDF	0.1 ppm	32 (1 ppm)	76	Less than 10% deviation after 500 bending cycles @ 60°	[11]
PAni-α-Fe_2_O_3_	PET	<2.5 ppm	72 (100 ppm)	50	-	[62]
GP-PAni	PVDF	100 ppb	60 (1 ppm)	46	The reponse decreased from 60 to 49% at 1500 bending cycles	[4]
PAni-WO_3_	PET	1 ppm	121 (100 ppm)	32	the response decreased by 9%@60° bending, 900 s	[58]
MWCNTs-PAni	Modified PET	33 ppm	117 (50 ppm)	47	-	[19]
PPy	silk	1 ppm	73.25 (100 ppm)	24	The response decreased by 10.61%@30° after 200 bending cycles	[63]
PAni-CoFe_2_O_4_	PET	25 ppb	118.3 (50 ppm)	24.3	Stable response after 500 bending cycle with 3.4% decrease@60°	This work

## Data Availability

Not Applicable.

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
