# Peer review of "A Novel Trace-Level Ammonia Gas Sensing Based on Flexible PAni-CoFe2O4 Nanocomposite Film at Room Temperature"

_polymers, 2021, doi:10.3390/polym13183077_

Round 1
Reviewer 1 Report
In this paper, the authors reported a trace-level ammonia sensor based on PANI-CoFe2O4 nanocomposite film. The authors have investigated the effects of CoFe2O4 on the gas sensing performances of gas sensor, and the results of NH3 sensing properties are acceptable. But there are some problems in research motivation, experimental details, and results. Manuscript may be accepted after major modification.
My specific comments are listed below:
- Introduction: (1) The reason for choosing CoFe2O4 as functional material is not enough. High remanence magnetization, coercivity, chemical stability, cost-effectiveness, and shape versatility are not the key reasons for selecting gas sensing material. (2) The authors ignored the review and discussion of PANI-based ammonia sensors. In fact, many PANI-based gas sensors have been reported. It is suggested to add corresponding discussions and references to highlight this work. Such as: Polymers 13 (2021) 1360; Sens. Actuators B 327 (2020) 128923. (3) It is suggested that references should be concentrated in the last three years.
- Experimental: The details about fabrication of CoFe2O4 should be given. Even though it has been referred to previous work.
- 2: How to get a constant relative humidity value (40% RH) in test chamber? Besides, gas distribution system is confusing as test gas is directly introduced into the chamber using one MFC. Furthermore, the balance gas of ammonia should be given, did you use the dry air or nitrogen as balance gas?
- 3: The full name of VSM should be given. Moreover, what conclusion about gas sensing properties can be obtained in VSM?
- 9: What conclusion about gas sensing properties can be obtained in TGA?
- 12d: How to obtain different relative humidity value in test chamber?
- The references in the Table 1 are insufficient and not new.
- Check the figures’ quality. Some figures have been deformed.
- The numbers in the chemical formula need subscripts, including table and references.
- English writing should be improved. Moreover, please carefully check the format/style of the target journal.
Reviewer 2 Report
This paper reports a novel trace-level ammonia gas sensing based on flexible PAni-CoFe2O4 nanocomposite film at room temperature. In general, the manuscript is well-organized, logically laid out and the experimental approach is technically sound. This paper is possibly publishable. For improving a manuscript, it is advisable to address the following comments:
- There are some typos in the manuscript. For example, ‘PANI’ should be ‘PAni’ in line 51 of Page 2. In line 126 of Page 4, ‘this is may be attributed’ should be ‘this is likely attributed’ or ‘this may be attributed’.
- What do red lines mean in the step of CoFe2O4 NPs in Figure 1?
- In Figure 2, please demonstrate how to control or change the various concentrations of ammonia gas in the test.
- In Figure 6a, it is not clear to see where you measured. In Figure 6d, it is better to use EDS mapping to verify that those two areas are CoFe2O4 NPs.
- Could the particle size of CoFe2O4 affect the sensitivity and limit of detection of this sensor? What is the optimal size?
- Please explain why not use higher content of CoFe2O4 (>50%) in the composite?
- In Figure 12 (c and e), is there any data treatment such as the baseline calibration after collecting the raw data?
- Will different HR levels affect the sensitivity and limit of detection of this sensor?
- Will the temperature affect the ammonia response and sensitivity of this sensor?
Round 2
Reviewer 1 Report
Concerns of reviewer have been addressed properly and publication is recommended.